# Single-Use Bioreactors for Human Pluripotent and Adult Stem Cells: Towards Regenerative Medicine Applications

**DOI:** 10.3390/bioengineering8050068

**Published:** 2021-05-17

**Authors:** Diogo E.S. Nogueira, Joaquim M.S. Cabral, Carlos A.V. Rodrigues

**Affiliations:** 1Department of Bioengineering and iBB—Institute for Bioengineering and Biosciences, Instituto Superior Técnico, Universidade de Lisboa, Av. Rovisco Pais, 1049-001 Lisboa, Portugal; diogoespiritosantonogueira@tecnico.ulisboa.pt (D.E.S.N.); joaquim.cabral@tecnico.ulisboa.pt (J.M.S.C.); 2Associate Laboratory i4HB—Institute for Health and Bioeconomy, Instituto Superior Técnico, Universidade de Lisboa, Av. Rovisco Pais, 1049-001 Lisboa, Portugal

**Keywords:** single-use bioreactors, regenerative medicine, human pluripotent stem cells, human mesenchymal stromal cells

## Abstract

Research on human stem cells, such as pluripotent stem cells and mesenchymal stromal cells, has shown much promise in their use for regenerative medicine approaches. However, their use in patients requires large-scale expansion systems while maintaining the quality of the cells. Due to their characteristics, bioreactors have been regarded as ideal platforms to harbour stem cell biomanufacturing at a large scale. Specifically, single-use bioreactors have been recommended by regulatory agencies due to reducing the risk of product contamination, and many different systems have already been developed. This review describes single-use bioreactor platforms which have been used for human stem cell expansion and differentiation, along with their comparison with reusable systems in the development of a stem cell bioprocess for clinical applications.

## 1. Introduction

The expression “regenerative medicine” dates back to at least 1992 [1], but its concept, the regeneration of tissues and cells damaged by ageing or disease, can be found as far back as Greek mythology [2,3]. Enraged by Prometheus’s betrayal, Zeus chained him to a rock and had his liver being eaten by an eagle every day, while it would fully regenerate every night. While regenerative medicine is not yet this efficient, landmark discoveries in stem cell biology and bioengineering in the past few years are leading to exciting advances that may revolutionise the field in the next decades.

Stem cells do not have a specialised function but, at the same time, are of crucial importance for human development and homeostasis. Stem cells can self-renew, generating identical copies of themselves upon division, and differentiate into specific, functional cells. The extent of cell types into which stem cells can differentiate depends on their potency.

Human pluripotent stem cells (hPSCs) can differentiate into any of the cell types comprising the human body (but not extraembryonic tissues). hPSCs include human embryonic stem cells (hESCs), which result from the in vitro culture of cells from the inner cell mass of the blastocyst [4], and human-induced pluripotent stem cells (hiPSCs), which are obtained by reprogramming of somatic cells [5,6]. Since the derivation of hESCs requires the destruction of human embryos, their use is forbidden in various countries. hiPSCs do not carry this ethical burden and can be derived from the patients’ own cells, thus overcoming the possibility of immune rejection. Besides regenerative medicine, hPSCs have been regarded as a promising source of cells for a variety of other applications such as drug screening and disease modelling [7], due to their ability to provide, in vitro, cells—or even structures—from diverse human systems and organs such as the heart, the central nervous system, or the liver [8,9].

The adult body also contains stem cells, responsible for the replacement/generation of specific cell types. Most of these stem cells are multipotent, generating only a limited number of different lineages. Concerning regenerative medicine applications, human mesenchymal stromal cells (hMSCs) are commonly regarded as promising candidates for cell therapies. These cells can differentiate into lineages such as bone, cartilage, and fat, and can be found in a variety of tissues, including the bone marrow (BM), umbilical cord matrix (UCM), adipose tissue (AT), peripheral blood, and synovial tissue [10]. hMSCs have been shown to be less immunogenic than other cells, allowing them to avoid adverse immune effects when transplanted to an allogeneic host, and also immunomodulatory, tuning and reducing the immune response of the host [11,12,13]. These characteristics have led to the exploration of hMSCs and their secretome for regenerative medicine applications (e.g., bone, cartilage, skin or trachea regeneration [14,15,16]), as well as for treatment and prevention of immune diseases, such as graft-versus-host disease, Crohn’s disease and multiple sclerosis [11,12,15]. hMSCs may also be promising for the treatment of patients afflicted with COVID-19, although some studies are still required on their safety and efficacy in this regard [17].

Nevertheless, despite the promise of both these types of stem cells, this promise can only be fulfilled if the cells can be expanded in a robust and reproducible manner to achieve the number of clinical-grade cells necessary for a patient. Effective hMSC doses are around 10^8^ cells/patient, with maximal effectiveness, in most clinical trials, with doses in the range of 70 million to 190 million cells/patient (intravenous administration requiring higher doses due to loss of some MSCs in the lungs) [18]. Point studies, however, have successfully applied higher dosages, including a phase 1b/2a clinical trial with 6.0 × 10^8^ MSCs/dose for Crohn’s disease [19] and a phase 2 clinical trial for acute ischaemic stroke using 1.2 × 10^9^ MSCs/dose [20]. In the case of hiPSC derivatives, one dose may constitute 10^9^–10^10^ cells [21]. Many stem cell bioprocesses, especially at lab scale, still employ planar culture platforms, including cell culture plates, T-flasks, or multi-plate trays. While cells produced in this manner may express normal markers and differentiation potential, as described in many studies, the 2D culture format affects their phenotype, altering surface marker localisation and sensitivity to signalling, as well as their behaviour, in terms of cellular processes such as expansion, differentiation, and apoptosis [22,23]. Therefore, these cells may not be of ideal quality for transplantation or for in vitro studies. Moreover, the increase in the scale of the vast majority of 2D platforms can only be performed using a scale-out approach—increasing the parallel number of culture vessels (plates, flasks, etc.)—rendering them unpractical or even unfeasible for the production of cells at a clinical scale.

Bioreactors have been long since established as promising platforms for the manufacturing of a number of different bioproducts [24,25,26,27] since they provide an agitated and homogenised 3D environment, as well as generally being available at many scales, allowing for scale-up approaches—increasing the size of the bioreactor itself. Furthermore, bioreactor vessels are typically equipped with probes, which can measure crucial culture variables such as pH, dissolved oxygen, and nutrient and metabolite concentrations (e.g., glucose and lactate, respectively), and which are associated with a controller system, which can react when these variables approach the limits of the established operating range, for instance, by adding acid/base, increasing aeration, changing the agitation speed or changing the culture medium flow rate. This know-how from traditional biological products has already been applied to stem cells, and various different studies already describe the expansion and/or differentiation of stem cells in different bioreactor configurations [28].

This review will detail the importance of single-use bioreactors in the production of stem cells for clinical applications, as well as explore some of the bioreactor systems which have already been described for the bioproduction of stem cells such as hPSCs and hMSCs.

## 2. Single-Use Bioreactors for Stem Cell Biomanufacturing

Despite all the advances in stem cell products for human use, regulation by agencies such as the European Medicines Agency (EMA) and the US Food and Drug Administration (FDA) is a critical concern. EudraLex (available online at https://ec.europa.eu/health/documents/eudralex, accessed on 11 April 2021) details the legislation for pharmaceutical products in the European Union, and Volume 4, in particular, details the guidelines for good manufacturing practices (GMP) for medicinal products for human and veterinary use. These GMP apply to various steps of the life of a stem cell product, from the cell bank establishment and maintenance to the manufacturing, downstream processing, fill, and finish (Figure 1). Among other things, GMP guidelines aim to avoid product contamination, thus requiring the sterilisation of the reagents and materials. Whenever possible, cleaning in place (CIP) and steaming in place (SIP) should be applied to sterilise materials between batches. The same regulation also permits and suggests the use of single-use technologies (EudraLex Volume 4, Part IV). In particular, the bioreactor parts which directly contact the cells can be disposable, which minimises the risk of contamination, and overrides the need for their sterilisation, thus reducing the time between batches and allowing a faster process pipeline and, thus, higher productivity.

The following sections will detail some single-use bioreactor platforms which have already been successfully applied in the biomanufacturing of stem cells. A summary of the results obtained in these bioreactors is presented in Table 1.

### 2.1. Single-Use Stirred-Tank Bioreactors

Stirred-tank bioreactors (STBRs) have been among the most commonly used setups for the manufacturing of a variety of products, such as viruses or recombinant proteins, in a multitude of cell types including bacterial, insect, plant, or animal cells [24,25,26,27]. STBRs typically employ a glass or stainless-steel vessel, along with one or multiple impellers distributed along the height of the bioreactor and ensuring efficient agitation of the medium (Figure 2). Depending on the frailty of the cells, different impellers can be employed—Rushton turbines are suitable for more resistant cell types, such as bacteria, while animal cells, which are more shear sensitive, require gentler mechanisms of agitation such as marine or pitched-blade impellers. In fact, shear stress is a common point of contention in the translation of stem cell culture to 3D, due to its effect on cell fate (reviewed in [72,73]). Nevertheless, there is a notorious lack of studies in this regard, with only a few reports of phenomena such as agitation-induced hPSC differentiation [74], as well as priming of hMSCs towards an osteogenic fate through high shear stress [75]. The decades of STBR use in the biopharmaceutical industry ensures these bioreactors are now very well characterised, in terms of agitation profiles, shear stress, power dissipation, and oxygen mass transfer, with well-established empirical correlations, as well as defined criteria for scale-up [76,77,78]. In STBRs, however, the existence of “hot spots” of high shear forces has been described as well as “dead zones” where mixing is inferior or almost inexistent. This heterogeneous mixing profile may give cues for undue cell differentiation and/or apoptosis due to the high shear itself or cell settling and formation of very large cell aggregates.

The STBR can be easily converted into a single-use system, by replacing parts which contact the cells with disposable equipment, such as the vessel itself, the impeller, and the probes. Examples of commercially available single-use STBRs which have already been successfully applied for stem cell culture are described in Table 2. Together, these systems cover from a 10 mL laboratory scale to a 200 L production scale.

Kropp et al. have described the expansion of hiPSCs as aggregates in BioBLU (from Eppendorf, Hamburg, Germany) bioreactors (125 mL working volume), using different feeding strategies. By applying a perfusion strategy, in which constant withdrawal of wasted medium and replenishment of fresh medium are performed (retaining the cells inside the vessel), the cell yield was improved by 47%, in comparison to a repeated batch approach, with a discrete medium exchange every 24 h. The authors obtained a final density of (2.85 ± 0.34) × 10^6^ cells∙mL^–1^, although with a shift to a metabolism more reliant on oxidative phosphorylation [29]. Kwok et al. expanded hiPSCs as aggregates in Mobius CellReady (from Merck, Darmstadt, Germany) bioreactors at a 1.5 L scale, generating a total of 2 billion cells [30]. hPSC differentiation in STBRs is also possible—Halloin et al. have adapted a standard cardiac differentiation protocol and report the generation of about 1 × 10^6^ cardiomyocytes∙mL^–1^ at both 100 mL and 350–500 mL scales, with purity above 90%. The bioreactors used in this study are available with both single-use and reusable vessels, which should allow for easy transition of protocols developed on one of those formats to the other [79].

Microcarrier-adherent hMSC expansion has also been described, in Mobius CellReady (from Merck, Darmstadt, Germany), UniVessel SU (from Sartorius, Göttingen, Germany), and CultiBag STR (from Sartorius, Göttingen, Germany) bioreactors. Schirmaier et al. optimised AT-derived hMSC culture at a 2 L scale, followed by a scale-up experiment to 35 L, and were able to maintain a similar cell density at both scales—(0.27 ± 0.02) × 10^6^ cells∙mL^–1^ and 0.31 × 10^6^ cells∙mL^−1^, respectively [34]. Lawson et al. attempted BM–hMSC culture at a 50 L scale, obtaining 1.28 × 10^10^ cells in 11 days of culture [36].

While obtaining high cell numbers is vital for the production of cells to be used in a clinical setting, bioprocess optimisation is preferably performed in a high-throughput platform. Ratcliffe et al. performed human haematopoietic stem/progenitor cell (hHSPC) expansion in the Ambr (from Sartorius, Göttingen, Germany) 15 bioreactor system, a fully automated and controlled platform entailing 24 vessels with 15 mL maximum volume, allowing for the simultaneous testing of many different conditions. By optimising culture parameters, including inoculation density, oxygen tension, and medium feeding, the authors could obtain a maximum density of 1.37 × 10^7^ cells∙mL^–1^ in 11 days [37]. This platform was also used for optimisation of BM–hMSC expansion with microcarriers in serum-free medium, resulting in over 8.1 × 10^5^ cells∙mL^−1^ and a 10-fold increase in reproducibility in comparison to serum-containing, manual culture [31].

### 2.2. Fixed-Bed Bioreactors

Fixed-bed bioreactors are a common staple of the food and wastewater industry due to their setup—a vessel filled with a macroporous material or large beads to which cells and/or enzymes may be attached, and through which the liquid phase is passed through continuously (Figure 3). In fact, many studies have already described the use of fixed-bed bioreactors for applications such as the production of biodiesel [80,81] and biohydrogen [82], and various aspects of wastewater treatment [83,84]. Beyond the possibility for these applications, fixed-bed bioreactors also have some characteristics which have led to their use for culture of anchorage-dependant mammalian cells. The geometry and setup of these bioreactors confer a large surface-to-volume ratio, allowing for a smaller footprint, compared to stirred-tank bioreactors, and enabling a much higher volumetric productivity. Depending on their mechanism, fixed-bed bioreactors may also be naturally compatible with a perfusion feeding regime, and, due to the lack of agitation, the shear stress conveyed to the cells is low. The lack of an impeller is, simultaneously, a disadvantage, due to allowing for the formation of concentration gradients (of nutrients, metabolites, growth factors, oxygen, etc.) inside the bioreactor. Moreover, cell harvesting until the end of the culture is impossible, rendering cell growth monitoring notably difficult. Nevertheless, fixed-bed bioreactors have seen used for applications such as viral production in mammalian cells [85,86,87], and even hHSPC expansion [88].

Single-use fixed-bed bioreactors require both a single-use vessel and cell adhesion matrix. Weber et al. have developed a small-scale disposable fixed-bed bioreactor system using a 3 mL plastic syringe, connected to two 250 mL flasks for medium feeding and for waste, and equipped with a small oxygen sensor in the outflow. This bioreactor system was used for the culture of alginate-encapsulated hMSC–TERTs. These cells are hMSCs transfected with telomerase, which compensates the telomere shortening which occurs during mitosis, thus enabling more population doublings in comparison to unaltered hMSCs [89]. These cells were not proliferative but could be maintained in culture for at least 500 hours with increasing viability [38]. The same group also described larger-scale fixed-bed bioreactor systems—using 60 mL glass syringes, and glass tubes between two stainless-steel plates (serving as lid and bottom) for a volume of 300 mL. These bioreactors were filled with 2 mm diameter non-porous borosilicate glass spheres which served as a surface for cell adhesion and made use of non-invasive oxygen sensors both at the inlet and the outflow. At the 300 mL scale, the authors were able to produce 6.2 × 10^8^ hMSC–TERTs after 167.3 hours of culture, however, only 50% to 60% of the inoculated cells attached to the borosilicate spheres [39]. Mizukami et al. have used a commercial fixed-bed bioreactor system—FibraStage^®^ bottles (discontinued) loaded with Fibra-Cel^®^ disks, both from Eppendorf (previously from New Brunswick Scientific) for the expansion of human cord blood-derived hMSCs. The bellows at the base of the bottle control the medium level by compression and expansion. With one 500 mL FibraStage bottle, the authors could produce (4.15 ± 0.81) × 10^8^ cells in 7 days of culture [40].

### 2.3. Hollow Fibre Bioreactors

Hollow fibre bioreactors are composed of numerous capillary tubes inside an outer shell, which, similarly to fixed-bed bioreactors, confer them a high surface area with a low footprint (Figure 4). In hollow fibre bioreactors, the cells adhere to the surface of the capillaries, either on the intracapillary (IC) or the extracapillary (EC) loop. Medium and reagents may be pumped through the loop in which the cells are adhered to, contacting them directly. The capillary membrane is semipermeable, thus also enabling the diffusion of dissolved gases and small molecules (e.g., glucose, lactate) through it, and allowing for mass exchange between the cells and the fluids perfused through the other loop [90]. Once more, the lack of an impeller reduces the shear stress conveyed to cells, especially due to the two-loop system and the possibility of indirect mass transfer. However, the growth of cells inside the hollow fibre bioreactors is hard to monitor due to the impossibility of cell harvesting until the end of culture. Having a membrane-based system also renders these bioreactors particularly susceptive to fouling—as the capillary membrane pores clog, either due to the cells or other solids, mass transfer through the capillaries becomes increasingly difficult [91]. The high cell densities which can be obtained in a hollow fibre reactor, along with the resulting high product titres, have led this configuration to be attractive for the production of recombinant proteins, monoclonal antibodies, and viruses [92,93,94] while also seeing the use for wastewater treatment [95,96] and biocatalytic reactions [97,98]. Furthermore, hollow fibre bioreactors have already been applied for the bioproduction of stem cells for clinical trials [43].

Regarding single-use hollow fibre bioreactors for stem cell expansion, many studies describe the use of the Quantum^®^ Cell Expansion System (QES; Terumo BCT). This closed and fully automated system is composed of about 11,500 hollow fibres, providing a 2.1 m^2^ surface area for cell growth (the same as 120 T-175 flasks) with a 0.3 m^2^ footprint.

Roberts et al. have harvested up to 5.4 × 10^8^ hESCs in 5 days in the QES, achieving 1.8-fold the cell density of T-25 flasks in an equivalent time, while reducing medium consumption per unit area in 68%, [41]. Mesquita et al. demonstrated hiPSCs could be expanded in the QES, and that laminin coating of the IC area was more favourable (both in terms of the maximum cell number and the time required to achieve it) for cell growth over vitronectin, possibly due to a stronger interaction of laminin with the QES fibres. Furthermore, to overcome some of the monitoring limitations of the bioreactor system, the authors correlated the total number of cells with the produced lactate, allowing them to estimate the growth curve of the hiPSCs. After 6–7 days of culture, the authors obtained an average of (6.9 ± 0.9) × 10^8^ cells with laminin coating [42]. Tirughana et al. described the production of GMP-grade human neural stem cells (hNSCs) in the QES. These hNSCs were modified to produce the prodrug-activating enzyme cytosine deaminase. A total of 1.4–3 × 10^9^ of these cells could be produced in one QES in 7–10 days, allowing for the production of an FDA-approved clinical lot (1.5 × 10^10^ cells) in seven parallel bioreactors in nine days, which was then used in a phase I trial with seven brain tumour patients. The same cell line was additionally modified by adenoviral transduction in the QES to produce a modified carboxylesterase, and 1.5–1.8 × 10^9^ cells could be recovered after 8 days. Using 5 QES in parallel, a clinical lot of 8 × 10^9^ transduced hNSCs was produced for a future study regarding the treatment of metastatic neuroblastoma patients [43].

The QES has already been extensively described for the culture of hMSCs of various sources – AT [45,48,51], BM [44,46,52], UCM [46,47,50], and periosteum [49]. Of these studies, Haack-Sørensen et al. were able to obtain the highest cell number. The authors performed a comparison of two media—foetal bovine serum (FBS) or human platelet lysate (hPL)-supplemented—for the derivation of hMSCs from an inoculum of the stromal vascular fraction of human subcutaneous abdominal fat, over the course of two passages in the QES. Overall, this study revealed hPL to be more favourable for cell growth. In the first passage, hPL increased cell yield by fivefold and in half of the expansion time when compared to FBS (9 days for hPL vs. 17 days for FBS). Both supplements led to comparable growth in the second passage, but with a higher difference in the expansion time (6 days for hPL vs. 21 days for FBS). FBS is an animal-derived supplement, which complicates the approval process in the case of a clinical-grade process. In contrast, hPL is of human origin, being free of xenogeneic components. Nevertheless, this supplement may also suffer from batch-to-batch variability [99]. At the end of culture, up to 5.5 × 10^8^ AT–MSCs were obtained in 9 days from stromal vascular fractions, and 6.1 × 10^8^ AT–MSCs were recovered after the first passage, following 6 days in culture [51]. Russell et al. have performed a cost breakdown, comparing automated hMSC expansion in the QES with manual culture for the production of 100 doses of 10^8^ cells (for a clinical trial with 100 patients). The automated system allowed for savings of about 49% in reagents and consumables, having an estimated cost of about USD 108,000 for 100 doses, while hands-on work could be reduced from 361.6 h (in a 24.0 week production time, thus requiring two technologists) to 35.0 hours (for a 20.0 week production, demanding a single technologist) [44]. Mizukami et al. performed a more thorough cost-of-goods analysis, comparing a multilayer vessel, a stirred-tank bioreactor, a packed-bed bioreactor, and a QES. While the QES resulted in a higher cell proliferation rate, expansion fold, and harvesting efficiency, its high consumable and equipment cost led it to be predicted as the least cost-effective option. In fact, for the QES to compete with the other systems at an economical level, it would require a sevenfold increase in harvesting density, along with an 85% consumable and equipment cost reduction and a 28% cost savings in the medium [47].

### 2.4. Rotary Cell Culture Systems

The advancements in space travel in the past years have led to significant research on the effects of microgravity in humans, and more specifically, in some cell types. While cell culture has already taken off to space [100], microgravity can also be simulated on Earth. The rotary cell culture system (RCCS) is a bioreactor developed by NASA which can be used to culture cells in a microgravity environment (Figure 5). In this bioreactor system, the cells are inoculated in a high aspect ratio cylindrical vessel, which is completely filled with medium. The vessel rotates horizontally and will cause the medium inside to rotate as well. If operating at a certain range of speeds, after some time, the medium will rotate at the same velocity as the vessel itself, unlike in other bioreactor systems where the fluid is moving in reference to the vessel walls (e.g., agitated liquid and still vessel walls). This minimises the effect of the Earth’s gravity on the particles inside the vessel, resulting in an effective gravitational force of 10^–2^ × *g* [101]. Moreover, the particles inside the vessel have an almost null terminal velocity and therefore move with the medium upon the rotational axis, with limited movement alongside other axes—a cell will return to approximately the same location upon each complete vessel rotation. Therefore, the cells can be cultured in suspension in a laminar regime; thus, with minimal shear stress (< 5 × 10^–2^ Pa), further reduced by the lack of headspace and, consequently, no formation of air bubbles [101,102,103]. Ensuring both these effects will require the rotation of the RCCS in a limited velocity range—a low rotation speed will not overcome the Earth’s gravitational force, causing the cells to settle, while high rotation will result in a predominant centrifugal force which will drive the cells towards the outer wall (Figure 5b–d). Furthermore, while the low shear stress conveyed by the RCCS limits the extent of shear-mediated cell damage, it also limits mass transfer by convection. In fact, in the RCCS, diffusive mass transfer prevails, which may lead to the formation of microenvironments of low nutrient concentration and high waste accumulation in the vicinity of the cells [103,104].

Single-use RCCS operate using disposable vessels, or cassettes, as these are the only part of the bioreactor which directly contacts the cells. Single-use RCCS systems are available commercially from Synthecon but only at a 10 mL and 50 mL scale, limiting their scalability.

Chiang et al. have employed the RCCS for hESC-derived hNSC culture and observed the effect of simulated microgravity in the development of these cells. RCCS culture led to increased β-adrenoreceptor expression and subsequent mitochondrial function of hNSCs, with increased mitochondrial mass and ATP production. Moreover, the proliferation of the hNSCs was increased, leading to a final density of ~5 × 10^5^ cells∙mL^–1^ after 3 days, a twofold increase over the static control [53]. The culture of adult BM–hMSCs in these bioreactors has also been described. Cells were harvested from BM and seeded in the RCCS in a chondrogenic medium. The results were compared with a standard protocol of cartilage production in conical tubes. After 2 weeks, RCCS culture led to about threefold larger (diameter-wise) and 10-fold heavier cellular constructs in relation to conical tubes, with a 1.5-fold higher glycosaminoglycan/DNA ratio and with histological and immunohistological characteristics of hyaline cartilage [54].

### 2.5. Rotating Bed Bioreactors

Rotating bed bioreactors apply rotation to the previously described fixed-bed system—the cells adhere to plates, which rotate inside the vessel (Figure 6). These bioreactors can be operated at full volume, or similarly to roller bottles, by not filling the vessel totally and allowing for the cells to intermittently contact the medium and the headspace. Rotating bed bioreactors share most advantages and disadvantages of fixed-bed bioreactors—both provide a large surface-to-volume ratio, are compatible with perfusion, and convey low shear stress to the cells, while cell harvesting and monitoring cannot be performed during the culture. Rotating bed bioreactors, however, can provide some more homogeneity to the contents of the vessel by means of rotation employed. Applications of these systems also include biocatalysis [105,106] and wastewater treatment [107,108], and it has already seen use in mammalian cell culture [109].

Zellwerk GmbH (Oberkrämer, Germany) has developed the disposable rotating bed Z^®^RP 2000 H and Z^®^RP 8000 H bioreactor systems, providing 2000 cm^2^ and 8000 cm^2^ of cell attachment area, respectively. These bioreactors consist of a cylindrical vessel containing a bed of polycarbonate plates which rotate through the action of a magnetic drive. Their setup allows for operation under a perfusion feeding regime. Moreover, they are equipped with a sampling device for supernatant analysis as well as pH and oxygen sensors connected to a control unit. The single-use parts comprise the vessel, tubing system, and measuring devices [55,56].

Neumann et al. characterised the Z^®^RP 2000 H bioreactor and applied it for UCM–hMSC expansion. The authors verified homogenisation of the culture under normal operating conditions for both a half-full (70 mL) and full (120 mL) vessel, and the Bodenstein number was characteristic of a stirred-tank reactor-type mixing even for the lowest mixing speed and full volume. After 5 days of culture, a total of (2.46 ± 0.24) × 10^7^ hMSCs could be obtained at a 125 mL scale, maintaining hMSC immunophenotype and trilineage differentiation potential [55]. Reichardt et al. applied rotating bed bioreactors with a 6000 cm^2^ surface area and a 340 mL working volume for human umbilical cord artery cells (HUCACs), harvesting (3.48 ± 0.55) × 10^8^ cells in 9 days, albeit with full depletion of glucose at some timepoints (despite the perfusion feeding regime). The authors compared bioreactor culture to maintenance in static T-25 flasks and estimated 311 of these flasks would be required to achieve the same cell numbers, encompassing an over threefold increase in the medium volume necessary (about 12.1 L versus 3.83 ± 0.69 L in the bioreactor) [56].

### 2.6. Rocking Motion Bioreactors

Rocking motion bioreactors were first described in 1998 [110] and were established as an alternative to STBR in the culture of insect and mammalian cells for processes such as the production of viruses and recombinant proteins. These bioreactors rely on an impeller-free agitation mechanism where the vessel, a plastic bag, is placed on the top of a base which moves in a back-to-forth rocking motion. The agitation causes the formation of waves in the air–liquid interface, ensuring efficient mixing, with a high mass transfer and no particle settling while also avoiding high shear stress (Figure 7) [111,112]. Cell damage is further avoided by the lack of a sparging mechanism and associated bubble formation. While the shear stress generally increases with the agitation velocity, at certain agitation speeds a resonance phenomenon is observed, causing unusual behaviour such as higher vorticity and shear stress. Zhan et al. observed that, for a half-filled 10 L rocking motion bioreactor, agitation at 15 rpm led to an approximation to the natural frequency of the bioreactor (calculated by approximating its shape to a perfect ellipse), causing higher turbulence than agitation at 22 rpm and 30 rpm [113]. Therefore, during process development with this bioreactor system, it is important to estimate the natural frequency and avoid choosing a rocking velocity close to this frequency.

Rocking motion bioreactors were first developed as single use, employing a disposable culture bag. Rocking motion bioreactors commercially available include the Xuri Cell Expansion System (previously WAVE Bioreactor^TM^) and Cellbag^TM^ vessels from Cytiva (Marlborough, MA, USA), and the Biostat^®^ RM and Flexisafe RM bags (replacing CultiBag) from Sartorius (Göttingen, Germany), and cover working volumes from 50 mL up to 500 L.

Davis et al. have cultured both hESCs and hiPSCs as aggregates in the Xuri Cell Expansion System, with perfusion and non-perfusion Cellbags, at various scales—150 mL, 250 mL, and 1 L—and also performed consecutive passage experiments. The authors also optimised the culture conditions and were able to obtain up to 9.5-fold expansion in 4 days. Serial passaging from 250 mL to 1 L under optimised conditions led to a 39.2-fold expansion (~3.9 × 10^9^ cells in 8 days), while passages from two sequential 150 mL bioreactors to 250 mL and to 1 L in non-optimised conditions resulted in a cumulative 280-fold increase in cell number (~1.7 × 10^10^ cells in 16 days). The authors also estimated how operating these bioreactors under automated perfusion, besides allowing a closed process and ensuring sterility, could lead to a substantial increase in cell yield per spent medium—from 4.2 × 10^5^ cells/mL in six-well plates to 1.5 × 10^6^ cells/mL in 1 L perfusion bioreactors [57].

hMSC expansion in rocking motion bioreactors has also been demonstrated by different groups. Nguyen et al. were able to expand these cells for 100 days in bioreactors with a 200 mL working volume, maintaining hMSC viability and chondrogenic and osteogenic differentiation potential. A maximum of (2.64 ± 0.18) × 10^8^ cells were counted on day 40. However, despite a decrease to (8.7 ± 2.1) × 10^7^ cells by day 90, the authors did not observe dead cells through a LIVE/DEAD assay and speculated the decrease in countable cell number to be attributed not to cell death but to hMSC migration to the inside of the microcarrier pores [58]. Da Silva et al. observed some cell deposition in Cellbags, which resulted in stagnated cell growth, and designed an acrylic grid which would allow for the Cellbag to remain in contact with the base for temperature control but would also raise the Cellbag wall, avoiding cell deposits. The authors also verified some deposits at the Cellbag wall due to the microcarriers sticking to it. The adhesion phase was first performed in spinner flasks, then in the modified Cellbags in dynamic conditions, and in static with reduced volume. The latter conditions led to the best overall cell expansion, with a final yield of 2.56 × 10^8^ UCM–hMSCs in 600 mL [59]. The same group then compared the expansion of these cells in the modified Cellbags (600 mL working volume) and 500 mL spinner flasks. Neither platform impacted hMSC tri-lineage differentiation potential nor differently regulated biological systems; however, spinner flask culture was more efficient, both in terms of final cell number and in the time at which the maximum was achieved. Maximum cell yields of (4.65 ± 0.14) × 10^8^ cells in 6 days and 1.32 × 10^8^ cells in 11 days were obtained for the spinner flask and the rocking motion bioreactor, respectively [60].

Timmins et al. have used rocking motion bioreactors for the production of red blood cells (RBCs) from umbilical cord blood (UCB)-derived hHSPCs. By controlling the cell density at regular intervals, the authors obtained an average 2.25 × 10^8^-fold increase in cell number in 33 days—a production averaging 4.5 × 10^15^ RBCs in a culture of up to 1 L of working volume and allowing to harvest 560 units of RBCs per UCB donation [61].

### 2.7. Vertical-Wheel Bioreactors

As mentioned in Section 2.1, stirred-tank bioreactors are the gold standard in traditional bioprocessing approaches but are undesirable for stem cell culture due to the high levels of shear stress their agitation mechanism conveys. While most other bioreactor systems already mentioned avoiding the issue by not employing an impeller, this impairs mass transfer and may compromise the homogeneity of the culture. The Vertical-Wheel^TM^ bioreactors (VWBRs), introduced by PBS Biotech (Camarillo, California, USA), were designed to provide a homogeneous culture environment while using a more gentle and efficient agitation mechanism (Figure 8) [114]. The vessels contain a large vertical wheel, which format results in both radial and axial agitation. Moreover, the U-shaped bottom of the vessel avoids “dead zones” underneath the impeller, thus limiting cell settling. Overall, this results in a more efficient mixing, allowing for low agitation speeds to be used and, consequently, less shear stress to be conveyed to the cells. Furthermore, the large size of the impeller allows for the dissipation of its rotational energy over a substantial area, thus resulting in narrower gradients of energy dissipation rate, with maximum values of 20–25%, compared to horizontal-blade impellers [114]. These bioreactors can have two different agitation mechanisms: in AirDrive VWBRs, agitation is controlled by bubble sparging, which are captured in specific zones of the vertical wheel, allowing for its motion; in MagDrive VWBRs, agitation is driven by magnetic coupling between magnets present in the vertical impeller and in the base unit. These bioreactors were developed to be single use and are available in various scales, from 100 mL to 80 L (MagDrive) or from 3 L to 500 L (AirDrive), and feature an embedded automatic controller starting from the 3 L units (Figure 8). The characterisation of VWBRs is so far limited, and therefore, no empirical correlations for bioreactor scale-up are yet available, but reports of computational fluid dynamics-based hydrodynamic modelling have already been published [66,114].

hiPSC culture in VWBRs has already been described, culturing the cells as aggregates (with a maximum yield of (7.1 ± 0.7) × 10^7^ cells in a repeated batch approach and (6.1 ± 0.7) × 10^7^ cells using a fed-batch process, both in 7 days and at a 60 mL scale [63]) and attached to vitronectin-coated microcarriers (maximum yield of (1.0 ± 0.1) × 10^8^ cells in 9 days at an 80 mL scale, and (2.6 ± 0.5) × 10^8^ in 8 days using a 300 mL working volume [64]). Notably, the addition of the polysulfated compound dextran sulphate, a staple of the biopharmaceutical industry due to aggregate size control via cell surface charge modulation and anti-apoptotic effect [115,116], improved the aggregate culture about twofold in terms of cell yield, leading to a maximum of (1.4 ± 0.1) × 10^8^ cells in 5 days with repeated batch and (1.26 ± 0.02) × 10^8^ cells in 6 days using a fed-batch process [63]. Borys et al. tested hiPSC expansion in 100 mL VWBR and found normoxic conditions using repeated batch to lead to a higher cell growth than experiments with hypoxia (3%) and/or batch conditions. By performing a low-density inoculation with pre-formed aggregates, they obtained up to (6.3 ± 0.4) × 10^8^ cells in 6 days, and about 2.1 × 10^12^ cells in 28 days, over the course of four consecutive passages [65]. The same group then optimised the culture protocol, in terms of inoculation (pre-formed aggregates vs. single cell), agitation speed and harvesting enzyme, and exposure time. With the optimised conditions, a maximum of (6.5 ± 0.6) × 10^8^ cells could be obtained, which could be passaged to 500 mL volume vessels [66]. In all cases, the expression of pluripotency markers and differentiation potential of the cells were found to be maintained. In a recent study, Silva et al. have demonstrated cerebellar differentiation of hiPSCs in VWBRs. The dynamic culture system was shown to maintain cell viability for at least 80 days of differentiation as well as to enhance extracellular matrix formation and to activate angiogenesis-related pathways in comparison to the static control [62]. This spontaneous onset of angiogenesis, in particular, is a very promising development since in vitro organoids are generally limited in this regard and necessitate alternative strategies for blood vessel formation [117].

Various studies also describe hMSC expansion in VWBRs. Yuan et al. developed a method for scalable BM–hMSC aggregation [67]. The authors designed thermal responsive poly(N-isporopylacrylamide) (PNIPAM) microcarriers to which cells could attach at 37 °C, and be detached from at the end of expansion by incubation at 23 °C. Following thermal detachment and microcarrier removal from the vessel, the cells would be left to spontaneously aggregate. The authors performed cell expansion, harvesting, and aggregation in both spinner flasks and VWBRs. In the spinner flask, the harvesting resulted in mainly single cells which did not aggregate even after 24 h. Expansion in the VWBRs at a 60 mL scale led to a production of 6.8 × 10^6^ cells in 4 days, which could be harvested in 10 min at 23 °C and could form aggregates, with a comparable diameter and cell viability to an AggreWell-based protocol, as well as with similar immunomodulation and cytokine secretion. Lembong et al. also describe BM–hMSC expansion in VWBRs, using a xeno-free, fed-batch approach. Following optimisation of the cell and microcarrier concentration, as well as the agitation, the fed-batch, and microcarrier addition strategies, and harvesting process (speed, quench hold time, and solution temperature) the authors could obtain 1.8–5.5 × 10^7^ cells across hMSCs of five different donors after 5 days in a 92 mL working volume [70]. Another report of xeno-free UCM– and AT–hMSCs expansion in VWBRs led to the production of (5.3 ± 0.4) × 10^7^ and (3.6 ± 0.7) × 10^7^ cells, respectively, in 7 days and with a 100 mL working volume. Furthermore, the VWBR was established as an economical alternative to T-175 flasks, as a cost analysis estimated a reduction of the process cost per dose, from USD 17,000 to USD 11,100 for UCM–hMSCs and from USD 21,500 to USD 11,100 for AT–hMSCs [68]. Similar conditions were used for extracellular vesicle (EV) production from BM–, AT– and UCM–hMSCs expanded in 100 mL working volume VWBRs. EVs comprise a prospective therapy for a variety of diseases, either by their own characteristics or as drug delivery vehicles. Up to (5.3 ± 5.5) × 10^7^ hMSCs were obtained after the culture (which ranged from 7 to 11 days depending on the cell donor and source). All three cell sources led to EV production in higher amounts when compared to static cultures (averaging 5.7 ± 0.9), with a maximum average of (6.9 ± 1.7) × 10^9^ particles/mL for UCM–hMSCs. The purity of these EVs was also improved and more consistent between runs in comparison to static culture [69]. Finally, BM–hMSCs have also been expanded at a 2.2 L scale, in AirDrive VWBRs. About 6.6 × 10^8^ cells could be obtained after 14 days of culture, with a similar cell concentration to 250 mL STBRs, although with a significantly lower percentage of apoptotic cells, as well as less HLA–DR expression (3% in VWBRs vs. 30% in STBRs). This lower HLA–DR expression, in particular, is promising for the development of allogenic cell therapies [71].

## 3. Challenges of Single-Use Bioreactor-Based Processes

The various examples of single-use bioreactors presented have shown that these platforms can provide an answer to the limitations of reusable bioreactor systems for the biomanufacturing of stem cells for regenerative medicine applications. In fact, disposable hollow fibre bioreactors have already been used in this context [43], and promising results on stem cell expansion and/or differentiation in other platforms, such as STBRs, rocking motion bioreactors, and VWBRs may lead into clinical trials with stem cells and derivatives produced in these bioreactors in the near future. The choice of bioreactor will depend on various factors, as each bioreactor has a subset of advantages and limitations (summarised in Table 3); however, the overall development of processes with single-use bioreactors also provides some challenges, independently of the platform of choice.

Reusable bioreactors necessitate resistant materials for their components, such as glass and/or stainless steel, in order to prolong their lifetime even under many successive/long cultures and various autoclaving or steaming cycles. The disposable parts of single-use bioreactors can employ less resistant materials, such as polymeric compounds, as they need only to be used for one culture process. However, their sterilisation with γ-radiation and their use under culture conditions can lead to the degradation of these polymers, which then leach to the culture medium. Marghitoiu et al. identified 53 different extractables from four different single-use vessels [118]. One of these, *bis*(2,4-di-*tert*-butylphenyl)phosphate (BdtBPP), is of special concern. BdtBPP results from the degradation of *tris*(2,4-di-*tert*-butylphenyl)phosphite (TBPP), an agent which is commonly added to polyolefins to protect them from oxidation by high temperatures or ionising radiation. BdtBPP was found to be a very potent inhibitor of cell growth [119,120,121], with a half-maximal effective concentration (EC_50_) for viable cell density between 0.12 mg/L and 0.73 mg/L across nine different CHO cell lines, while up to 2 mg/L could be extracted from single-use vessels [120]. Notably, as the ageing of single-use vessels leads to a reduction in the leachable compounds, some vessels where cell growth was impaired by the presence of BdtBPP could sustain normal cell growth after some time [122]. Nevertheless, due to the possibility of leaching-mediated impaired cell growth, extensive tests on the vessel extractables and care in the use of compounds such as TBPP are of crucial importance. The lesser resistance of the single-use bioreactors in comparison to reusable systems may also limit the scale at which cultures can be performed—at very high scales, the vessels may not be able to withstand the liquid volume and rupture. While this is not a problem for autologous therapies, especially if high-density cultures can be performed, the establishment of off-the-shelf products may be more difficult.

Single-use bioreactors also present an important sustainability issue. In fact, the disposal of possibly hundreds or thousands of single-use vessels over the lifetime of a process will contribute to a higher footprint since these vessels, due to having harboured expanding cells, are treated as biological waste and cannot be recycled, being incinerated, or sent to landfills. The manufacturing, packaging, sterilisation, and shipping of these bioreactors also pose sustainability constraints, as these steps have to be constantly repeated throughout the process. However, a set of life cycle analyses of monoclonal antibody and adenoviral vaccine production in reusable and single-use systems actually concluded the single-use process to be more environmentally sustainable. In fact, the major environmental impact of traditional systems lies in the sterilisation, specifically, on the water for injection and clean steam requirements. Conversely, the end of life of disposable systems was found not to be significant in comparison to the impact of both their production and of the bioprocess operation. Therefore, overall, single-use bioreactors were considered by the authors a “greener” approach, but their environmental impact can be reduced even further by, for instance, ensuring the bioproduction facility is as close as possible to the site of vessel manufacturing, and by promoting the recycling of any non-contaminated parts, namely, the packages and wrapping [123]. Furthermore, this study focused on traditional biological processes, as such, it still remains unanswered if the same conclusions can be extrapolated to the bioproduction of stem cells with its associated specificities.

The transition to disposable systems is also of economic concern [124,125]. While disposable vessels are less expensive than their reusable counterparts, resulting in a smaller initial investment, the need for new vessels for each bioreactor run may lead to higher running costs throughout the lifetime of the process. The differences in single-use bioreactor process cost will depend on the lower initial investment and higher consumable cost, in addition to other factors such as the time saved in sterilisation cycles and the potential lower risk of contaminated batches. Furthermore, processes with single-use bioreactors may face smaller regulatory hurdles, reducing the time-to-market and, consequently, the pay-back and pay-out times. The evaluation of the most financially attractive process will require a thorough economic analysis. Even with this analysis, the choice will depend on what the final goals are—for instance, to obtain a higher profit in the lifetime of the process or to break even with the initial costs at the earliest opportunity.

Regardless of the type of bioreactor used, the development of a stem cell biomanufacturing process benefits from a quality by design (QbD) framework. This approach allows for the definition of the design space in which the process can operate, allowing for some flexibility in the process while ensuring the final product quality. Furthermore, quality control is shifted upstream in the process timeline—instead of being applied at the product level, the whole process is subjected to quality control, allowing for early detection should problems arise. If the characteristics of each single-use vessel are included as critical process parameters, QbD may allow mitigating the effect of possible vessel-to-vessel variations. QbD is also fuelled by continuous and iterative improvement—as more insight on the process is gathered, it can be modified to account for this new information [126]. Therefore, QbD is an important tool for the development of a robust bioprocess while reducing expenses on product quality control and failed batches and allowing for further steps towards meeting GMP guidelines.

## 4. Conclusions

Human stem cell-based therapies have gained much traction in recent years. However, the myriad of issues raised by the large-scale manufacturing processes of stem cells and their derivatives—such as the limited knowledge about the complexity of human stem cells; the impact of the dynamic culture environment provided by bioreactors on cell phenotype, viability, or quality; the prohibitive cost of platforms and reagents; the shift in paradigm from cells as a factory to cells as the final product, and the regulatory hurdles associated with this shift—have slowed their transition to the clinic. Furthermore, cell differentiation, which is required in many cases (namely, in all processes involving hPSCs), increases the complexity of the processes and the various studies already performed in 2D may be difficult to translate to a bioreactor environment. Nevertheless, some of these barriers are being breached, and it is already possible to produce clinical-scale numbers of stem cells without compromising their quality. In fact, most reported studies using bioreactors to culture stem cells and derivatives perform evaluations of cell phenotype as well as genomic integrity. However, more comprehensive studies (e.g., proteomic or transcriptomic analysis of static vs. dynamic cultures [62]) are still necessary to fully understand the effect of the dynamic culture microenvironment present in bioreactors on the identity of the generated cells, namely envisaging their clinical use. While earlier reports employed reusable bioreactor systems, the various types of single-use bioreactors here reviewed have also been established as competent platforms to ensure hPSC and hMSC expansion and differentiation. Furthermore, single-use bioreactors reduce the risk of contamination of the final product, thus allowing for a more GMP-compliant process. As processes with single-use systems develop and more insight is gathered on how to answer their limitations, they can be expected to take an important role in the establishment of stem cell products, thus allowing for more steps to be taken towards regenerative medicine being more than just a promise, and Prometheus’s regenerating liver more than just a myth.

## Figures and Tables

**Figure 1 bioengineering-08-00068-f001:**
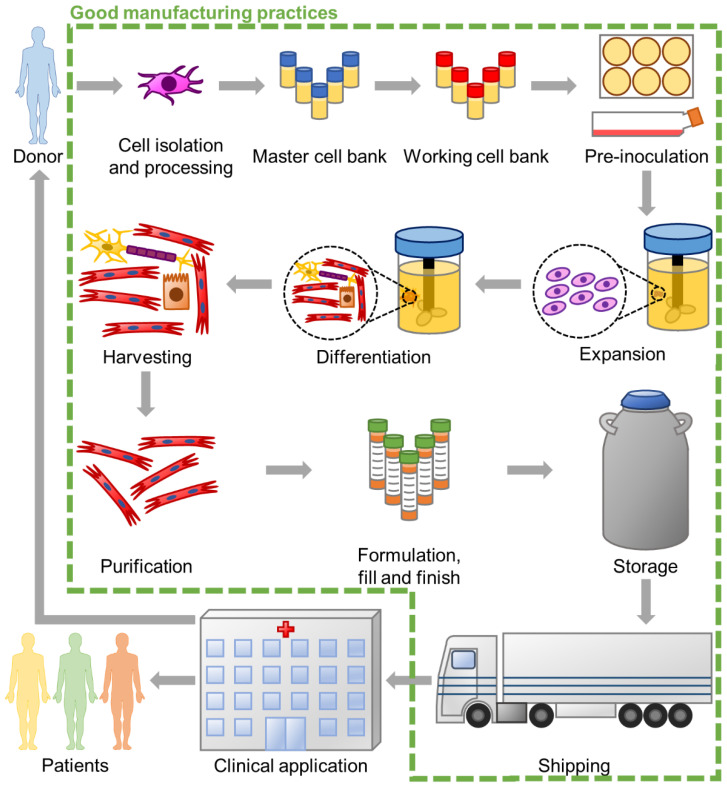
Process pipeline for the production and clinical application of a stem cell product (either autologous or allogeneic). European Medicines Agency legislation established good manufacturing practices to be applied from the initial cell isolation and processing to the product fill and finish while also requiring a clear definition of the storage and shipping conditions of the finished cell product. We note that the figure depicts a general stem cell product pipeline and, although most processes already at clinical scale do not perform yet differentiation in the bioreactors. However, we believe the field will move in that direction since using planar platforms will be hardly feasible at a clinical scale.

**Figure 2 bioengineering-08-00068-f002:**
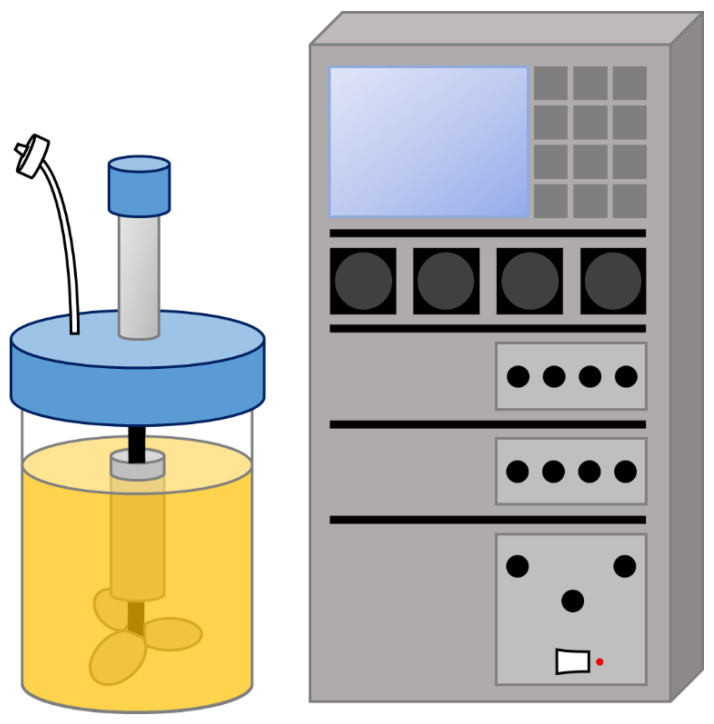
Schematics of a stirred-tank bioreactor vessel and controller unit.

**Figure 3 bioengineering-08-00068-f003:**
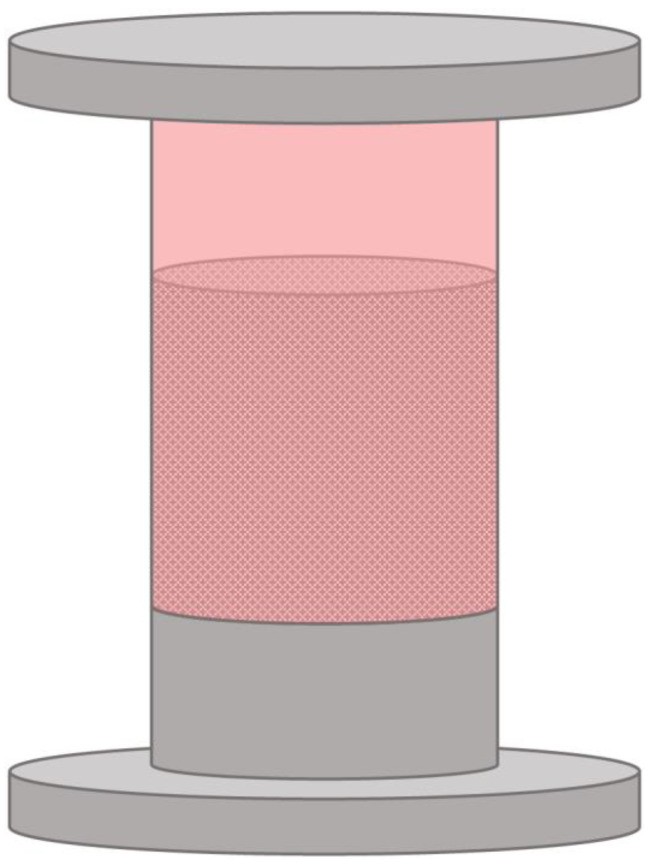
Schematics of a fixed-bed bioreactor.

**Figure 4 bioengineering-08-00068-f004:**
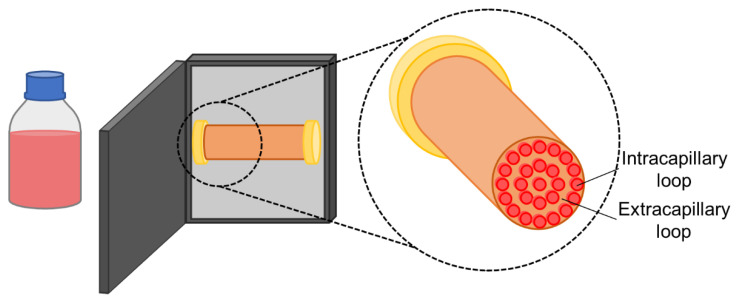
Schematics of a hollow fibre bioreactor system and a close-up of the hollow fibre module.

**Figure 5 bioengineering-08-00068-f005:**
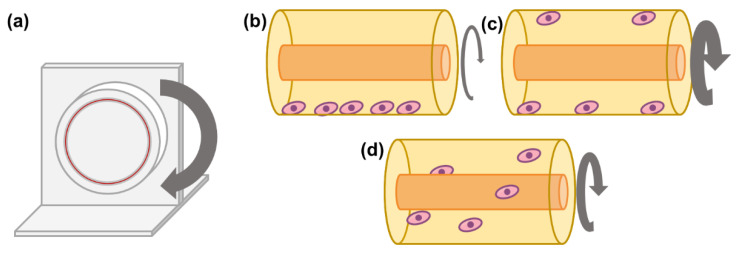
Schematics of a rotary cell culture system (RCCS): (**a**) four-station rotator base with one 50 mL vessel and two 10 mL vessels; (**b**) at low rotation speeds, the cells will settle along the bottom of the vessel; (**c**) at very high rotational speeds, the cells will be subjected to a predominant centrifugal force, driving them towards the outer wall of the vessel; and (**d**) at a certain velocity range, the cells will be in suspension, in a simulated microgravity environment.

**Figure 6 bioengineering-08-00068-f006:**
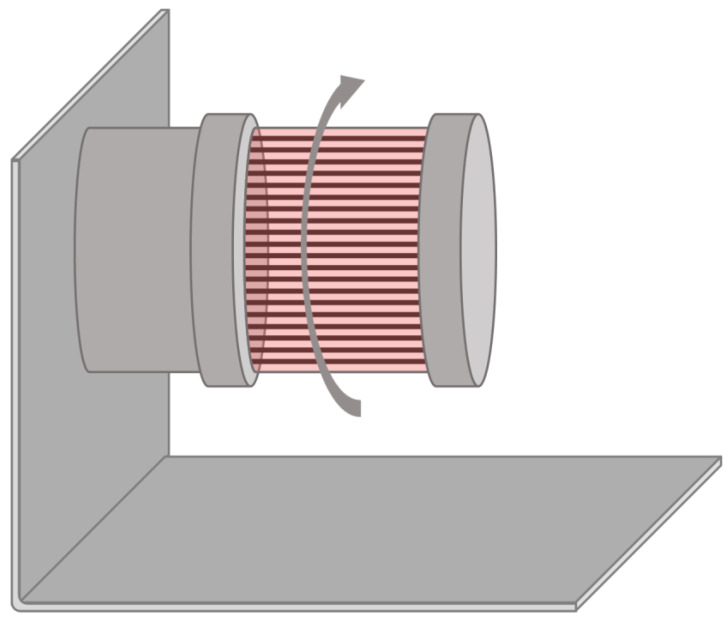
Schematics of a rotating bed bioreactor.

**Figure 7 bioengineering-08-00068-f007:**
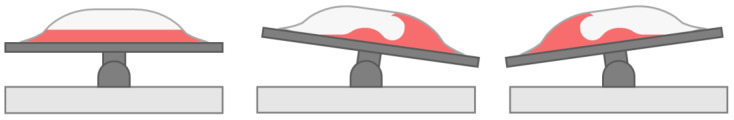
Schematics of a rocking motion bioreactor. The back-and-forth rocking motion of the base will lead to the formation of waves inside the vessel, allowing for efficient mixing.

**Figure 8 bioengineering-08-00068-f008:**
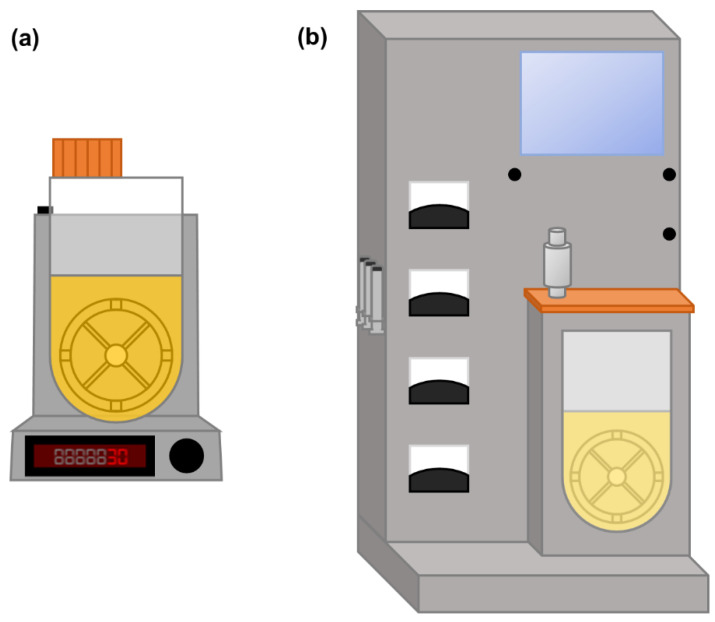
Schematics of a Vertical-Wheel bioreactor: (**a**) 100 mL vessel with base and (**b**) 3 L vessel with an embedded controller.

**Table 1 bioengineering-08-00068-t001:** Comparison of single-use bioreactor systems for stem cell culture. Working volume ranges indicate a change in volume during the culture. Final cell densities presented are the maximum average obtained among different conditions and/or donors: hESC—human embryonic stem cell; hHSPC—human haematopoietic stem/progenitor cell; hiPSC—human induced pluripotent stem cell; hMSC—human mesenchymal stromal cell; hNSC—human neural stem cell; RBC—red blood cell.

Bioreactor Type	Cell Type	Working Volume/Area	Culture Time (days)	Maximum Final Cell Density	Ref.
Stirred tank	hiPSCs	125 mL	7	(2.9 ± 0.3) × 10 ^6^ cells∙mL^–1^	[29]
		1.0-1.5 L	7	(1.99 ± 0.09) × 10 ^6^ cells∙mL^–1^	[30]
	hMSCs	15 mL	8	8.1 × 10 ^5^ cells∙mL^–1^	[31]
		100–200 mL	10	1.8 × 10 ^5^ cells∙mL^–1^	[32]
		1.0–2.0 L	7	4.1 × 10 ^5^ cells∙mL^–1^	[33]
		2.0 L	7	(2.7 ± 0.2) × 10 ^5^ cells∙mL^–1^	[34]
		1.0–2.4 L	14	~ 1 × 10 ^5^ cells∙mL^–1^	[35]
		35 L	7	3.1 × 10 ^5^ cells∙mL^–1^	[34]
		50 L	11	2.6 × 10 ^5^ cells∙mL^–1^	[36]
	hHSPCs	10 mL	10	1.4 × 10 ^7^ cells∙mL^–1^	[37]
Fixed bed	hMSCs	3 mL	20.8	N/A ^(1)^	[38]
		14.2 mL	5.6	(2.9 ± 0.1) × 10 ^6^ cells∙mL^–1^	[39]
		60 mL	7.0	1.75 × 10 ^6^ cells∙mL^–1^	[39]
		300 mL	6.9	2.05 × 10 ^6^ cells∙mL^–1^	[39]
		500 mL	7	(8.3 ± 1.6) × 10 ^5^ cells∙mL^–1^	[40]
Hollow fibre	hESCs	2.1 m^2^	5	3.4 × 10 ^4^ cells∙cm^–2^	[41]
	hiPSCs	2.1 m^2^	6–7	(3.3 ± 0.4) × 10 ^4^ cells∙cm^–2^	[42]
	hNSCs	2.1 m^2^	7–11	1.5 × 10 ^5^ cells∙cm^–2^	[43]
	hMSCs	2.1 m^2^	7–9	N/A ^(2)^	[44]
		2.1 m^2^	17 ± 6	(4.7 ± 0.6) × 10 ^3^ cells∙cm^–2^	[45]
		2.1 m^2^	8 ± 2	(8.0 ± 2.5) × 10 ^3^ cells∙cm^–2^	[46]
		2.1 m^2^	5	(9.8 ± 1.0) × 10 ^3^ cells∙cm^–2^	[47]
		2.1 m^2^	5	(1.1 ± 0.2) × 10 ^4^ cells∙cm^–2^	[48]
		2.1 m^2^	7.9–9.9	(1.8 ± 0.2) × 10 ^4^ cells∙cm^–2^	[49]
		2.1 m^2^	6	(1.9 ± 0.3) × 10 ^4^ cells∙cm^–2^	[50]
		2.1 m^2^	6	2.9 × 10 ^4^ cells∙cm^–2^	[51]
		2.1 m^2^	6–13	4.7 × 10 ^4^ cells∙cm^–2^	[52]
Rotary cell culture system	hNSCs	4 mL	3	~ 5 × 10 ^5^ cells∙mL^–1^	[53]
	hMSCs	10 mL	14	N/A ^(1)^	[54]
Rotating bed	hMSCs	2000 cm^2^	5	(1.2 ± 0.1) × 10 ^4^ cells∙cm^–2^	[55]
		6000 cm^2^	9	(5.8 ± 0.9) × 10 ^4^ cells∙cm^–2^	[56]
Rocking motion	hESCs	150 mL ^(3)^	4	2.8 × 10 ^6^ cells∙mL^–1^	[57]
		400 mL ^(3)^	4	1.4 × 10 ^6^ cells∙mL^–1^	[57]
		1.0 L ^(3)^	4	1.3 × 10 ^6^ cells∙mL^–1^	[57]
	hMSCs	50–200 mL	100	(1.32 ± 0.09) × 10 ^6^ cells∙mL^–1^	[58]
		50–600 mL	10	4.4 × 10 ^4^ cells∙mL^–1^	[59]
		50–600 mL	11	2.2 × 10 ^5^ cells∙mL^–1^	[60]
	hHSPC-RBCs	200 mL–1 L	33	4.5 × 10 ^12^ cells∙mL^–1^	[61]
Vertical-Wheel	hiPSCs	60 mL	80	N/A ^(1)^	[62]
		60 mL	7	(2.3 ± 0.2) × 10 ^6^ cells∙mL^–1^	[63]
		60–73 mL	7	(1.79 ± 0.03) × 10 ^6^ cells∙mL^–1^	[63]
		80 mL	9	(1.21 ± 0.02) × 10 ^6^ cells∙mL^–1^	[64]
		300 mL	8	(8.6 ± 1.5) × 10 ^5^ cells∙mL^–1^	[64]
		100 mL	6	(6.3 ± 0.4) × 10 ^5^ cells∙mL^–1^	[65]
		100 mL	6	(6.5 ± 0.6) × 10 ^5^ cells∙mL^–1^	[66]
		500 mL	6	~4 × 10 ^5^ cells∙mL^–1^	[66]
	hMSCs	60 mL	4	1.1 × 10 ^5^ cells∙mL^–1^	[67]
		60–100 mL	7	(5.3 ± 0.4) × 10 ^5^ cells∙mL^–1^	[68]
		60–100 mL	7–11	5.3 × 10 ^5^ cells∙mL^–1^	[69]
		90–92 mL	5	~ 6 × 10 ^5^ cells∙mL^–1^	[70]
		2.2 L	14	~ 3 × 10 ^5^ cells∙mL^–1^	[71]

^(1)^ Non-proliferative and/or differentiating cells. ^(2)^ The exact cell density achieved is not stated by the authors but is inferred from the text to be around 10^5^ cells∙cm^–2^. ^(3)^ Part of a consecutive passage experiment with increasing scale; overall fold increase of ~280 over 16 days. Individual experiments at each scale with optimised conditions led to better results, but the exact cell density is not indicated in the article.

**Table 2 bioengineering-08-00068-t002:** Examples of single-use stirred-tank bioreactor systems with successful use for stem cell culture.

Bioreactor	Company	Impeller	Working Volume Range
BioBLU^®^	Eppendorf	Eight-blade or pitched-blade	100 mL–40 L
Mobius^®^ CellReady	Merck	Marine (scoping)	1.0–2.4 L
Ambr^®^	Sartorius Stedim Biotech	Pitched-blade or Rushton	10–250 mL
BIOSTAT^®^ CultiBag STR Plus	Sartorius Stedim Biotech	Three- or six-blade	12.5–200 L
UniVessel^®^ SU	Sartorius Stedim Biotech	Three-blade	600 mL–2.0 L

**Table 3 bioengineering-08-00068-t003:** Advantages and limitations of single-use bioreactors as whole and specific single-use platforms.

Platform	Advantages	Drawbacks/Limitations
Single-use bioreactors	Compatible with GMP guidelines Pre-sterilised—no CIP and SIP necessary Closed systems—minor contamination risk Reduced downtime and higher productivity Lower overall environmental impact than reusable systems Lower initial investment	Risk of leachables—possible cell growth impairment Maximum scale limited by material resistance Environmental impact of vessel manufacturing, packaging, shipping, and disposal throughout the whole process High running costs
Stirred tank	Vast know-how and characterisation Available at many different scales Availability of empirical correlations and criteria for variable estimation and scale-up Variety of agitation mechanisms Some are naturally compatible with perfusion	High overall shear stress Heterogeneity of shear stress distribution—existence of hot-spots and stagnated zones
Fixed bed	Low shear stress High surface-to-volume ratio and small footprint Naturally compatible with perfusion	Formation of concentration gradients Cell harvesting only possible at the end of the culture Difficult cell monitoring
Hollow fibre	Low shear stress High surface-to-volume ratio and small footprint Semipermeable membrane system, allowing for indirect mass exchange Naturally compatible with perfusion	Formation of concentration gradients Cell harvesting only possible at the end of the culture Difficult cell monitoring Susceptibility to fouling Expensive operation Available only at a single scale (2.1 m^2^)
Rotary cell culture system	Low shear stress Simulated microgravity environment No air bubbles Some are naturally compatible with perfusion	Formation of concentration gradients Available only at low scales (up to 50 mL)
Rotating bed	Low shear stress High surface-to-volume ratio and small footprint Intermittent contact with medium and headspace Naturally compatible with perfusion	Cell harvesting only possible at the end of the culture Difficult cell monitoring
Rocking motion	Efficient mixing with low shear stress No air bubbles Some are naturally compatible with perfusion Available at many different scales	Resonance phenomenon—spike of shear stress at certain rocking velocities Some cell deposition and microcarrier sticking to vessel walls
Vertical-Wheel	Efficient mixing with low shear stress Vessel format avoids cell settling beneath the impeller Narrow gradients of energy dissipation rate Available at many different scales Naturally compatible with perfusion starting from the 3 L scale	Still not well characterised Small-scale (100 mL and 500 mL) bioreactors not controlled and incompatible with perfusion

## Data Availability

Not applicable.

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
