# Peer review of "Single-Use Bioreactors for Human Pluripotent and Adult Stem Cells: Towards Regenerative Medicine Applications"

_bioengineering, 2021, doi:10.3390/bioengineering8050068_

Round 1

Reviewer 1 Report

General Comments

This is an extremely well written review that manages to present a complex combination of biological and engineering principles in an easy to read manner.  The authors have organized a diverse body of papers into concise and logical presentation.  The work incorporates not only scientific and engineering concepts for the cost-efficiency and regulatory issues that will drive manufacturing efforts.  The information is presented without any evidence of commercial influence or bias.  This reviewer has only one suggestion:

  1. The authors are in a unique position to elaborate more precisely on the effect of shear forces on stem cell growth, differentiation and viability.  Since these parameters are the topic of investigation by multiple manuscripts covered in this review, it would be instructive to summarize the range of shear forces observed in the context of the different cell lineages.  Specifically, can the authors provide a table that summarizes the upper and lower shear force limits compatible with the different cell lineages covered in this review?  Having these parameters identified in a single review paper could be helpful to the literature as a whole.  There is clearly a need for these types of summary tables in the emerging field of Microphysiological system (MPS) perfusion bioreactor papers!

Specific Comments:

Ln 283.  Spell out $108K as $108,000.

Ln 333.  Provide an area or volume basis for the number of expanded cells reported for the device, not simply an absolute number of cells as currently presented.

Ln 489.  Provide a numbered reference at the end of the sentence with regard to the Yuan et al. citation mentioned in the sentence.

Author Response

Reviewer appreciation: “This is an extremely well written review that manages to present a complex combination of biological and engineering principles in an easy to read manner.  The authors have organized a diverse body of papers into concise and logical presentation.  The work incorporates not only scientific and engineering concepts for the cost-efficiency and regulatory issues that will drive manufacturing efforts.  The information is presented without any evidence of commercial influence or bias.  This reviewer has only one suggestion:”

Author response: We thank the reviewer for their appreciation of our work and for the time and attention given for its review.

Reviewer comment #1: “The authors are in a unique position to elaborate more precisely on the effect of shear forces on stem cell growth, differentiation and viability.  Since these parameters are the topic of investigation by multiple manuscripts covered in this review, it would be instructive to summarize the range of shear forces observed in the context of the different cell lineages.  Specifically, can the authors provide a table that summarizes the upper and lower shear force limits compatible with the different cell lineages covered in this review?  Having these parameters identified in a single review paper could be helpful to the literature as a whole.  There is clearly a need for these types of summary tables in the emerging field of Microphysiological system (MPS) perfusion bioreactor papers!”

Author response #1: We thank the reviewer for their comment and share the opinion that such a summary table would greatly benefit the scientific community in this regard. Unfortunately, to the best of our knowledge, most studies do not report the range of shear forces present in the reported bioreactor cultures, presenting instead the stirring speed which is not a comparable parameter between different studies. Indeed, the same stirring speed with different impeller geometries may lead to very different results in terms of shear forces. Moreover, studies on the exact effect of shear stress on the cells (namely, shear stress values which induce their differentiation or death) are in very limited scope and, as such, we do not have enough information to construct a table as suggested by the reviewer. Still, we added the following comment to the manuscript:

Line 146: “In fact, shear stress is a common point of contention in the translation of stem cell culture to 3D, due to its effect in cell fate (reviewed in [72,73]). Nevertheless, there is a notorious lack of studies in this regard, with only a few reports of phenomena such as agitation-induced hPSC differentiation [74], as well as priming of hMSCs towards an osteo-genic fate through high shear stress [75].”

Reviewer comment #2:

“Ln 283.  Spell out $108K as $108,000.

Ln 333.  Provide an area or volume basis for the number of expanded cells reported for the device, not simply an absolute number of cells as currently presented.

Ln 489.  Provide a numbered reference at the end of the sentence with regard to the Yuan et al. citation mentioned in the sentence.”

Author response #2: We thank the reviewer for recognising these issues and have corrected them accordingly.

Regarding the second comment, the sentence was altered to:

“Moreover, the proliferation of the hNSCs was increased, leading to a final density of ~ 5 × 105 cells∙mL–1 after 3 days, a 2-fold increase over the static control [53].”

Reviewer 2 Report

In this review, Nogueira D.E.S et al. aim at describing single-use bioreactors that have been used with human stem cells and their possible use for clinical studies. The review is thorough and well documented addressing the diverse bioreactors used for stem cells expansion. However, a few points remain elusive and should be addressed:

  • The review principally focuses on pluripotent cells expansion, and often, more specifically MSCs. Given the highly controversial use of MSC in clinic and lack of consensus in their regards, caution is required when addressing their clinical translation. For example, in the introduction, line 61-62, the promise of hMSCs for COVID-19 treatments should be toned down. Line 65-66, the numbers of hMSCs “doses” should be specified according to some pathologies/studies.
  • In figure 1, the GMP application seems to apply only to the cells production process, but cell storage, and shipping should also be included in the GMP box.
    Also, cells are being differentiated within the bioreactors which is almost not reported in the different bioreactors’ descriptions, or not at a clinical scale. This could be a bit misleading.
  • Similarly, and this should clearly be discussed, the fact that bioreactors can fairly easily be upscaled for cells production, it is clearly much more complicated for differentiated 3D bioreactor tissue culture. In fact, reference 52 in part 2.4 line 340 refers to a fairly old study (2009) which led to tissues still far from clinical scale. It makes it look like there was no significant improvement since then.
  • In table 3, the pros and cons of bioreactors as a whole could be visually (e.g. double lines, colors…) separated from the rest for clarity.
  • The discussion on the environmental impact should also be toned down as is the conclusion of ref. 114, or more references going in this direction should be added. Ref 115 on the other hand should be removed as it is based on a book chapter itself referring to a non-peer reviewed magazine promoting technological solutions.
  • A reference for economic concern paragraph (line 575-586) would be welcome.

Overall, this review would be suitable for publication in bioengineering given the points mentioned above would be addressed.

Author Response

Reviewer appreciation: “In this review, Nogueira D.E.S et al. aim at describing single-use bioreactors that have been used with human stem cells and their possible use for clinical studies. The review is thorough and well documented addressing the diverse bioreactors used for stem cells expansion. However, a few points remain elusive and should be addressed:”

Author response: We thank the reviewer for their appreciation of our work and for the time and attention given for its review.

Reviewer comment #1: “The review principally focuses on pluripotent cells expansion, and often, more specifically MSCs. Given the highly controversial use of MSC in clinic and lack of consensus in their regards, caution is required when addressing their clinical translation. For example, in the introduction, line 61-62, the promise of hMSCs for COVID-19 treatments should be toned down. Line 65-66, the numbers of hMSCs “doses” should be specified according to some pathologies/studies.”

Author response #1: We thank the reviewer for their comment. We realise hMSCs are controversial, especially when regarding such a recent and still not fully understood disease such as COVID-19. We toned down the comment regarding hMSC prospective use for COVID-19 and we included more detail regarding the numbers of hMSCs per dose required:

Line 66: “Effective hMSC doses are around 108 cells/patient, with maximal effectiveness, in most clinical trials, with doses in the range of 70 million to 190 million cells/patient (intravenous administration requiring higher doses due to loss of some MSCs in the lungs) [18]. Point studies, however, have successfully applied higher dosages, including a phase 1b/2a clinical trial with 6.0 × 108 MSCs/dose for Crohn’s disease [19] and a phase 2 clinical trial for acute ischaemic stroke using 1.2 × 109 MSCs/dose [20].”

Reviewer comment #2: “In figure 1, the GMP application seems to apply only to the cells production process, but cell storage, and shipping should also be included in the GMP box.”

Author response #2: We thank the reviewer for their comment. We only included the steps directly related to the cell production process in the GMP box as per the European law (the regulations for GMP production of advanced therapy medicinal products available at https://ec.europa.eu/health/sites/default/files/files/eudralex/vol-4/2017_11_22_guidelines_gmp_for_atmps.pdf). However, the same regulation does require the manufacturer to specify conditions for the cell product storage and shipping. As such, we included shipping and storage in the GMP box and edited the figure legend accordingly.

Reviewer comment #3: “Also, cells are being differentiated within the bioreactors which is almost not reported in the different bioreactors’ descriptions, or not at a clinical scale. This could be a bit misleading.”

Author response #3: We thank the reviewer for this comment and recognise their concern regarding the limited studies regarding cell differentiation. While undifferentiated hMSCs can be used for in vivo applications (although with some controversy, as the reviewer pointed out), hPSCs have to be differentiated before being used in a clinical setting. Furthermore, we believe that biomanufacturing of differentiated cell products at a clinical scale will require the differentiation to be performed in bioreactors, as cell differentiation in planar culture platforms at that scale is close to unfeasible, while also causing cellular alterations (as discussed in the manuscript). We added this note to the legend of Figure 1:

We note that the figure depicts a general stem cell product pipeline and, although most processes already at clinical scale do not perform yet differentiation in the bioreactors. However, we believe the field will move towards that direction as using planar platforms will be hardly feasible at a clinical scale.”

Reviewer comment #4: “Similarly, and this should clearly be discussed, the fact that bioreactors can fairly easily be upscaled for cells production, it is clearly much more complicated for differentiated 3D bioreactor tissue culture. In fact, reference 52 in part 2.4 line 340 refers to a fairly old study (2009) which led to tissues still far from clinical scale. It makes it look like there was no significant improvement since then.”

Author response #4: We acknowledge the reviewer’s suggestion and added the following comment to the conclusions:

Line 641: “Furthermore, cell differentiation, which is required in many cases (namely in all processes involving hPSCs), increases the complexity of the processes and the various studies already performed in 2D may be difficult to translate to a bioreactor environment.”

We have also added two recent references to the manuscript, one describing cerebellar differentiation of hiPSCs in Vertical-Wheel bioreactors and another with cardiac differentiation of hiPSCs in stirred tank bioreactors (despite not being clear if the bioreactors used are single-use or reusable, the fact they are available on both formats should allow for transition of the protocol from one format to the other – this comment was also added to the manuscript):

Line 510: “In a recent study, Silva and colleagues have demonstrated cerebellar differentiation of hiPSCs in VWBRs. The dynamic culture system was shown not only to maintain cell viability for at least 80 days of differentiation, but also both to enhance extracellular matrix formation and to activate angiogenesis-related pathways in comparison to the static control [62]. This spontaneous onset of angiogenesis, in particular, is a very promising development, as in vitro organoids are generally limited in this regard and necessitate alter-native strategies for blood vessel formation [117].”

Line 175: “hPSC differentiation in STBRs is also possible – Halloin and co-workers have adapted a standard cardiac differentiation protocol and report the generation of about 1 × 106 cardiomyocytes∙mL–1 at both 100 mL and 350-500 mL scales, with purity above 90%. The bioreactors used in this study are available with both single-use and reusable vessels, which should allow for easy transition of protocols developed on one of those formats to the other [79].”

Reviewer comment #5: “In table 3, the pros and cons of bioreactors as a whole could be visually (e.g. double lines, colors…) separated from the rest for clarity.”

Author response #5: We thank the reviewer for their comment and recognise the lack of clarity in Table 3. We have opted for separating the pros and cons with bullets to improve readability.

Reviewer comment #6: “The discussion on the environmental impact should also be toned down as is the conclusion of ref. 114, or more references going in this direction should be added. Ref 115 on the other hand should be removed as it is based on a book chapter itself referring to a non-peer reviewed magazine promoting technological solutions.”

Author response #6: We acknowledge the reviewer for this comment. We have removed reference 115, and due to a lack of references regarding this point (to our knowledge), we have maintained the conclusions of reference 114 but toned them down accordingly.

Reviewer comment #7: “A reference for economic concern paragraph (line 575-586) would be welcome.”

Author response #7: We acknowledge the reviewer for this comment. Once more, to our knowledge, there are a lack of studies regarding the economic concerns of single-use and reusable systems, as such, the discussion was based on the authors’ view without presenting any system as superior. Nevertheless, as per the reviewer’s suggestion, we have added two references (references 124 and 125) to this paragraph which could be of interest to the readers.

Reviewer 3 Report

This is an excellent review which covers an important area which has not been brought together in this way previously to my knowledge. The breadth and literature is good with pros and cons of the different type of bioreactors well considered. The review blends research activity with commercial instruments well.

A weakness perhaps is not fully considering those systems which are close to clinical use and the potential effects on phenotype for these clinical applications could be better described. An ex would be the Quantum towards Autologous chondrocyte implantation which is being considered clinically.

Author Response

Reviewer appreciation: “This is an excellent review which covers an important area which has not been brought together in this way previously to my knowledge. The breadth and literature is good with pros and cons of the different type of bioreactors well considered. The review blends research activity with commercial instruments well.”

Author response: We thank the reviewer for their appreciation of our work and for the time and attention given for its review.

Reviewer comment: “A weakness perhaps is not fully considering those systems which are close to clinical use and the potential effects on phenotype for these clinical applications could be better described. An ex would be the Quantum towards Autologous chondrocyte implantation which is being considered clinically.”

Author response: We thank the reviewer for their suggestion. We further evidenced the fact that the Quantum Cell Expansion System has seen use in at least one clinical trial and added the following comment, referencing stirred tank, rocking motion and Vertical-Wheel bioreactors, as we feel they might be closer to clinical applications due to their scalability and compatibility with cell growth under different formats (e.g. aggregates, microcarriers and encapsulated) which confers them more flexibility:

Line 555: “In fact, disposable hollow fibre bioreactors have already been used in this context [43], and promising results on stem cell expansion and/or differentiation in other platforms, such as STBRs, rocking motion bioreactors and VWBRs may lead into clinical trials with stem cells and derivatives produced in these bioreactors in the near future.”

Regarding the phenotype of the bioreactor cultured cells, we agree with the reviewer that this is a critical point. Likewise, we added this sentence to the manuscript:

Line 646: “In fact, most reported studies using bioreactors to culture stem cells and derivatives perform evaluations of cell phenotype as well as genomic integrity. However, more comprehensive studies (e.g. proteomic or transcriptomic analysis of static vs dynamic cultures [62]) are still necessary to fully understand the effect of the dynamic culture microenvironment present in bioreactors on the identity of the generated cells, namely envisaging their clinical use.”